# How Do Type 2 Diabetes Patients Value Urban Integrated Primary Care in China? Results of a Discrete Choice Experiment

**DOI:** 10.3390/ijerph17010117

**Published:** 2019-12-23

**Authors:** Xin Wang, Kuimeng Song, Paiyi Zhu, Pim Valentijn, Yixiang Huang, Stephen Birch

**Affiliations:** 1School of Public Health, Health Development Research Center, Sun Yat-Sen University, Guangzhou 510080, China; wangxin25@mail.sysu.edu.cn (X.W.); zhupy3@mail2.sysu.edu.cn (P.Z.); 2Shandong Institute of Medicine and Health Information, Shandong First Medical University & Shandong Academy of Medical Sciences, Jinan 250012, China; miramax@126.com; 3Department of Health Services Research, Care and Public Health Research Institute (CAPHRI), Faculty of Health, Medicine and Life Sciences, Maastricht University, 6229 GT Maastricht, The Netherlands; valentijn@essenburgh.nl; 4Integrated Care Evaluation, Essenburgh Research & Consultancy, 3849 AE Hierden, The Netherlands; 5Centre for the Business and Economics of Health, University of Queensland, Brisbane, QLD 4072, Australia; stephen.birch@uq.edu.au

**Keywords:** primary care, discrete choice experiment, type 2 diabetes, preference

## Abstract

*Objectives:* Fragmented healthcare in China cannot meet the needs of the growing number of type 2 diabetes patients. The World Health Organization proposed an integrated primary care approach to address the needs of patients with chronic conditions. This study aims to measure type 2 diabetes patients’ preferences for urban integrated primary care in China. *Methods:* A discrete choice experiment was designed to measure type 2 diabetes patient preferences for seven priority attributes of integrated care. A two-stage sampling survey of 307 type 2 diabetes mellitus (T2DM) patients in 16 community health stations was carried out. Interviews were conducted to explore the reasons underpinning the preferences. A logit regression model was used to estimate patients’ willingness to pay and to analyze the expected impact of potential policy changes. *Results:* Travel time to care providers and experience of care providers are the most valued attributes for respondents rather than out-of-pocket cost. Attention to personal situation, the attentiveness of care providers, and the friendliness and helpfulness of staff were all related to interpersonal communication between patients and health care providers. Accurate health information and multidisciplinary care were less important attributes. *Conclusions:* The study provides an insight into type 2 diabetes patients’ needs and preferences of integrated primary care. People-centered interventions, such as increasing coverage by family doctor and cultivating mutual continuous relationships appear to be key priorities of policy and practice in China.

## 1. Introduction

The past 30 years have witnessed significant increases in the prevalence of type 2 diabetes mellitus (T2DM) in China, increasing from 0.67% in 1980 [1] to 11.6% in 2013 [2,3]. In 2016, T2DM and its complications led to 1.46% total deaths, 2.39% disability-adjusted life years (DALYs) and 3.96% years lived with disability (YLDs) in the population of China [4]. The estimated 110 million T2DM patients require access to coordinated, ongoing, and organized care delivered by teams of skilled healthcare providers [5,6]. However, the Chinese health system has been characterized by a disease-focused hospital-centered approach with a weak primary care system since the 1980s [7,8]. The core functions of primary care in prevention, case detection and management, gatekeeping, referral, and care coordination for T2DM are not being met, resulting in a high undiagnosed rate, delayed diagnoses and treatment, and complications of T2DM. For instance, a nationally representative cross-sectional survey showed that the awareness, treatment, and control rates for diabetes were 36.5%, 32.2%, and 15.84%, respectively [9]. Another study reported a proportion of undiagnosed adult diabetes patients of 70% [10], while the World Bank found that admission rates for complications from diabetes in China are more than five times the rate in OECD countries, which is an indicator of poor primary health care [11].

These problems indicate a need for innovative solutions in Chinese health systems, particularly for strengthening diabetes-related care delivery through a primary care approach [6]. Since the new round of health reform in 2009, the government has undertaken to “Equalize public health services for all”, by funding primary care providers to deliver a defined package of basic public health services, including management of diabetes [8,12]. The management of T2DM patients consists of one fasting blood glucose test and two postprandial blood glucose tests per year in the high-risk population, four face-to-face follow ups and one physical examination of each patient per year, together with health education of the whole population, mainly focusing on prevention and detection [13]. Although a study in 2012 showed that newly diagnosed patients increased over a five-years period, the life-course management and care coordination of T2DM were still absent [14]. In the last decade, integrated care for diabetes and other chronic conditions has been recognized as a means to meet the needs of this population by promoting care coordination, improving the quality of care, and generating better patient outcomes in developed countries [15,16]. In 2016, the global report on diabetes by the World Health Organization (WHO) proposed that integrated health services could deliver a continuum of health promotion, prevention, diagnosis, treatment, rehabilitation, and palliative care through the different levels and sites of care within the health systems over the course of a diabetes patient’s lifetime, and suggested orienting integrated care around people’s needs [6]. Hence, integrated care in the primary care system is a strategy for meeting patient needs, reducing the burden of disease, and improving the management of T2DM.

However, careful design of such a reform will depend on understanding patient preferences for integrated care in Chinese primary care settings. Discrete choice experiments (DCEs), provide an important source of patient preference and can be used to explore whether patients are willing to trade-off some of the attributes of health care against the others [17]. DCEs have been used to elicit and quantify preferences in primary care models, health plans, vaccination, and employment of health personnel in developed countries [18,19,20]. With regard to integrated care, only two similar studies have been conducted in the US and Germany to analyze patient preferences for integrated care provided in the whole health system [21,22]. However, the results illustrate preference differences in the two different systems. Due to differences of cultural backgrounds and health system contexts, previous results are not generalizable to the Chinese setting. The aim of this study is to elicit the preferences regarding the use of urban integrated primary care in China using a DCE and interviews with T2DM patients.

## 2. Materials and Methods

### 2.1. Study Setting

Nanshan is one of the ten districts in Shenzhen city, with an area of 182 km^2^ and 1.4246 million people. Shenzhen has been a leader in developing integrated healthcare systems in China [23,24]. There is a city-level hospital, five district-level hospitals, and 80 community health stations (CHSs) in Nanshan. The prevalence of T2DM in Shenzhen was 6.2% of the entire population in 2013, with 48.49% of them unaware. In 2015, there were 112,000 people over 60 years old in Nanshan, accounting for 8.67% of the whole population. However, it is estimated that the number of the elderly is increasing by 6.52% yearly [25,26,27]. The 80 CHSs received 3.52 million outpatient visits, accounting for only 34.26% of all outpatient visits in the district in 2017, indicating that patients are bypassing CHSs for their care needs.

The demands of patients with chronic conditions for continuous healthcare is increasing. Given the increase of the T2DM patients, together with population aging, the Health and Family Planning Commission (HFPC) in Nanshan realized that it is necessary to shift the hospital-centered and treatment focused healthcare system to a more community-based integrated care system. This study will elicit T2DM patient preferences for integrated care in CHSs to promote the shift. 

### 2.2. DCE Design

#### 2.2.1. Attributes and Levels

The first step in a DCE is to select the attributes and levels to be included, which determines the validity of the DCE [28]. Based on a review of the literature, we adopted the conceptual framework for integrated care of Mühlbacher and colleagues [29]. As shown in Figure 1, it consists of 21 attributes under seven dimensions, incorporating individual-level, procedural-level, and organizational-level features.

In order to choose the attributes and levels representing realistic characteristics in contemporary Chinese primary care systems and reduce respondent burden, we conducted key informant interviews (*n* = 12) with healthcare experts in China and two focus group discussions each with 20 T2DM patients. Appropriateness of the attributes for the Chinese primary care context was scored using a three-point Likert scale. Respondents could add comments regarding their score. In addition, “Out of pocket costs” was included as the 8th dimension, allowing for the estimation of patients’ willingness to pay for improvements in other attributes. Table 1 lists the final attributes and corresponding levels. According to the geographical distribution of CHSs in Nanshan district, the three levels of travel time to care provider we identified are 10 min, 20 min, and 30 min. Through a literature review about cost of diabetes patient per visit [30] and two key informant interviews with GPs working in community health stations of Nanshan district, we defined the three levels of “out of pocket costs”, ¥10, ¥20, and ¥30 per visit.

#### 2.2.2. Questionnaire

A full factorial design of eight attributes (with three levels each) generates 3^8^ = 6567 alternatives. We used 27 alternatives, which is more manageable, using a fractional factorial experimental design [31]. The alternative with the most attributes at the middle level was selected as a comparator and the other 26 alternatives were compared to it, to make the choices easier to understand for respondents [32]. Each choice set consisted of two community health station alternatives, alternative A (the comparator) and alternative B, for which attribute levels were varied systematically. Thereafter, we conducted 12 pretest interviews with diabetes patients, which demonstrated that they felt comfortable with the simple and patient-friendly language, they understood the choice sets, and they could make logical trade-offs between alternative A and B. To avoid overloading the respondents, we randomly divided the 26 choice sets into three different questionnaires, with two questionnaires containing nine choice sets each and one questionnaire containing eight choice sets.

Each questionnaire contained two parts. Part one consisted of an introduction to the study topic, informed consent, and questions about sociodemographic information and previous experience of illness and seeking care. Part two included a choice set example including an explanation of the term used, and the choice sets. 

#### 2.2.3. Sampling and Data Collection

A two-stage sampling process was followed. The first stage sampled CHSs, and the second stage sampled T2DM patients. Sixteen of 80 community health stations (20%) were selected by systematic sampling. In each CHS, 20 patients with T2DM were randomly selected from the list of residents who registered with GP in the CHS. The survey was conducted in April 2018. Participation in the survey was voluntary, and the participants were not compensated. Trained interviewers guided the respondents to complete the questionnaire. In addition, a researcher interviewed patients and staff in the CHSs to explore reasons underlying the choices made. The study (No.2017-035) complied with the recommendation of the Declaration of Helsinki, and was approved by the Ethical Committee of School of Public Health, SUN Yat-Sen University.

### 2.3. Data Analysis

We used Stata 15 to analyze the DCE data. The basis of DCE is the random utility model.

In the model, patient *n* is assumed to be a rational economic individual who faces a choice among I alternative CHSs in each of T choice sets, and n will choose CHS alternative i over alternative j if and only if $U_{ni}$ ≥ $U_{nj}$. The utility that patient *n* receives from alternative I in choice set t is
(1)$${⋃}_{nit}=\beta_{n}^{\prime }x_{nit}+\varepsilon_{nit}$$

The personal utility of an alternative is not directly observable. $x_{nit}$ is a vector containing the attributes of alternatives, while coefficient vector $\beta_{n}$ is unobserved for each *n* and varies in the population with density f($\beta_{n}|\theta^{*}$) where $\theta^{*}$ represents the true parameters of this distribution and $\varepsilon_{nit}$ is an unobserved random term. It is assumed that $\varepsilon_{nit}$ is independent and identically distributed (iid) extreme value. Conditional on $\beta_{n}$, the probability that patient n chooses alternative i in choice set t is given by the standard logit:(2)$$L_{nit}(\beta_{n})=\frac{\exp\left(\beta_{n}^{\prime }x_{nit}\prime \right)}{\sum_{j=1}^{i}\exp\left(\beta_{n}^{\prime }x_{n_{j}t}\right)}$$

The clogit model was used in this study.

#### 2.3.1. Willingness to Pay

The estimated coefficients in the regression give information about the direction and significance of the effects of changing the levels of one attribute, but it does not provide a valuation necessary for comparison of alternative policies. We therefore calculate willingness to pay (*WTP*) and policy impact measures based on regression results using Stata’s nlcom command.

In this study, the *WTP* to get a higher level of a specific healthcare attribute in CHSs is measured as the willingness to pay more out-of-pocket cost. As the out-of-pocket cost per visit is continuous and quadratic, the *WTP* for attribute x can be estimated as:(3)$$WTP\left(x\right)=\frac{\partial U/\partial x}{\partial U/\partial c_{ni}}$$
where $c_{ni}$ represents out of pocket cost. The *WTP* measure is calculated for all out-of-pocket cost per visit levels, but we only report the highest out-of-pocket cost per visit level as it best reflects the current status in Nanshan district.

#### 2.3.2. Impact of Policy Change

Policy makers are interested in how the probability of a patient choosing a given CHS changes as levels of attributes are changed. Because the $\varepsilon_{ni}$ is not given, the choice probability is the integral of $P_{ni}|\varepsilon_{ni}$ over all values of $\varepsilon_{ni}$ weighted by its density. 

The logit probability of choosing alternative i rather than alternative *j* is given by:(4)$$P_{ni}=\frac{e^{\alpha_{1}c_{ni}+\alpha_{2}c_{ni}+\beta^{\prime }x_{ni}}}{\sum_{j}e^{\alpha_{1}c_{nj}+\alpha_{2}c_{nj}+\beta^{\prime }x_{nj}}}$$
where x is a vector of attribute coefficient. If the policy in CHS i is changed, the impact of policy is the change in the probability of seeking healthcare in baseline CHS k:(5)$$\mathrm{Impact}\;\mathrm{of}\;\mathrm{policy}\;\mathrm{change}=P_{ni}-P_{nk},\;i\neq k$$
where k is the baseline CHS and i is the CHS with policy changes.

## 3. Results

### 3.1. Respondents’ Characteristics

The final sample used in the analysis consisted of 307 respondents and 5318 observations, with each respondent providing response to eight or nine choices. The response rate was 95.9%. Additionally, 10 diabetes patients and four staff in CHSs were interviewed. Table 2 shows the demographic and health care characteristics of the sample. Most respondents were married, with a mean age of 63 years old, and a quarter of the respondents were employed. Just over 57% of all respondents were enrolled in a health insurance plan of Shenzhen city, which was much lower than the national average level (i.e., Shenzhen is an immigrant city). A quarter of the respondents have diabetes complications. The mean frequency of healthcare visits was 1.2 visits/month, while only 1/3 respondents have registered with a GP in CHSs and reported “very good, good or fair” levels of health.

### 3.2. Preferences for CHS Attributes and Willingness to Pay

Table 3 presents the clogit regression results. In the regression, all reference categories are the lowest levels of each attribute. Except for multidisciplinary care and accurate health information, all coefficients are statistically significant. T2DM patients prefer the nearest CHS, in which providers with more experience give more attention to patients, and the providers are friendly and helpful when patients have questions or problems. The coefficient between the second and third level of multidisciplinary care, and the coefficient between the second and third level of accurate health information are negative, but they are not statistically significant.

Table 3 also presents the willingness to pay results, which explain how much in out-of-pocket costs a T2DM patient is willing to pay to get a higher level of healthcare provided by CHSs. T2DM patients on average are willing to pay the largest amount of money for “Travel time to care providers—10 min” (¥40.18), and followed by “Experience of care provider—over 10 years” (¥37.72), “Attention to personal situation—always considered” (¥23.08), and “Attentiveness of care providers—always given” (¥21.89). There is no statistically significant willingness to pay more out-of-pocket cost for improving the level of “Multidisciplinary care” and “Accurate health information”. To sum, the results show that location is by far the most valued feature of selecting a CHS among sampled T2DM patients. Attributes that are related to care providers, such as the working experience, attitude and behavior (patient-centered), are also very important for T2DM patients. Accessing health information and coordination among care providers appear not to be as important.

### 3.3. Impact of Policy Change

We forecasted the effectiveness of different policies by the changes in the probability of choosing the baseline CHS. The baseline CHS was set at level 3 of all seven attributes.

Table 4 shows that raising the working experience of care providers in CHSs to 10 years is the most efficient way to attract T2DM patients visiting CHSs. It will increase the probability of a T2DM patient visiting CHS by 91.7%. Convenient location of CHSs has the second largest impact on the probability of T2DM patients visiting the CHS. Reducing the travel time to CHS from 30 min to 20 min could increase the probability of visiting CHS by 79.1%. Reducing the travel time to CHS from 30 min to 10 min could increase the probability by 89.9%. Increasing care providers’ time helping patients understand their health problems and care plans and considering patients work, social, and family situations for finding the most appropriate care plan, could also increase the changes in probability dramatically. It should be noted that raising the frequency of care providers’ friendliness and helpfulness from “sometimes” to “always” could increase the probability by 72.25%, but raising it from “sometimes” to “often” is not helpful for attracting patients. Patients have a high requirement of friendliness and helpfulness for care providers. When reducing the out-of-pocket cost from ¥30 to ¥20 and gradually to ¥10, each reduction steadily increased the probability of visiting CHSs by around 19%. This indicates that, compared to capacity of care provider and people centeredness, out-of-pocket cost has less influence on T2DM patients’ choices.

## 4. Discussion

This study determined the importance T2DM patients assign to each attribute of integrated care in urban primary care systems and their willingness to pay, and calculated how choice probabilities vary with changes in attribute levels by DCE. As patients’ choice of receiving healthcare is represented as a multicriteria decision, our findings will be important for modeling of integrated care for T2DM and other chronic conditions in the primary care systems of China.

### 4.1. Interpretation of the Results

Travel time to care providers is the most decisive healthcare characteristic for respondents in Nanshan. The result is also confirmed by interviews with respondents. “I always go to the Majialong community health station. Go down this street and turn right, you will find it, next to a vegetable market. I usually grab some medicine on my way back from the vegetable market…” Not only less travel time, but also located close to places frequented by the elderly could attract respondents to CHSs. HFPC of Nanshan district are aware of this issue and have adopted a “10 min healthcare circle” for building new CHSs and relocating some of the existing CHSs in the whole district since 2017. 

The care provider’s experience is the second most important attribute for respondents. This could reflect perceptions that work experience can not only improve care providers’ capacity of providing appropriate health care, but also contributes positively to patient trust. Some researchers found that providers’ capacity for providing care and quality of care influenced patients’ choice of CHS [33]. Data provided by the HFPC of Nanshan district showed that 43.7% (141/323) of staff who quit from CHSs had more than five years of working experience, and 49.4% (206/417) of staff who were newly hired by CHSs had less than five years of working experience during 2014–2018. (Nanshan HFPC, 2018, unpublished data) Taking human resource cost into consideration, it may be optimal for the HFPC of Nanshan district to hire staff with a few years working experience and reduce the turnover rate of former employees in CHSs. Additionally, encouraging endocrinologists with rich experience from district hospitals of vertical medical consortia to set up clinics regularly in CHSs [34] would not only improve capacity of CHSs, but also promote interorganizational care coordination.

A few researchers have found that attentiveness, friendliness, and helpfulness of staff affect patients’ choices [35]. We also identified patients’ preferred levels of the people-centered attributes. The lack of gatekeepers and care coordination resulted in low levels of continuous care between patients and providers in China, as well as low levels of attentiveness and attention from providers. In interviews with respondents, we found that most of them have little understanding of what people-centered care is, and they seem not to be enthusiastic about participating in developing care plans (average 6.5 score out of total 10 score). However, patients’ needs and expectations are central features of integrated care [16,36]. Increasing coverage by family doctors, cultivating mutual continuous relationships, and strengthening training of staff in human relations may represent cost-effective policies for CHSs to attract T2DM patients. Care pathways might emphasize friendliness and helpfulness throughout the patient experience together with more attention and attentiveness to patients.

Accurate health information and multidisciplinary care were considered less important by respondents. Some respondents did not know how to deal with health information by himself/herself. “I don’t know how to access the health information with my mobile phone. Even if I get the health information, I can’t read it.” Moreover, most CHSs are technically unable to share information with other CHSs or patients in Nanshan district, and integrated care has not been provided by multidisciplinary teams in Nanshan. One staff member in Nanyou CHS said “We have been working on encouraging residents to sign contract with GP since 2016. But the residents seem to be not interested in it because there is no difference before and after contracts, we provide the same healthcare by the same means. There are three so-called multidisciplinary teams in our CHSs, but we do not practice as a team.” Although these two attributes are not priorities for T2DM patients in Nanshan, researchers in high-income countries demonstrate that health information is the basis of health system improvements [37]. Clinical integration, which is based on multidisciplinary care teams, is the key for providing integrated care [38]. Multidisciplinary care was the second highest priority in patients’ choice in Germany [21]. Therefore, it might be beneficial for the HFPC of Nanshan district to design a comprehensive patient-based health information system and promote multidisciplinary care delivery in advance.

### 4.2. Strengths and Limitations

This study contributes to the existing literature and practice two ways. First, the realistic attributes and levels of integrated care in CHSs of China were identified by a combination of literature review, key informant interviews, and focus group discussions. The identification will contribute to awareness and understanding of integrated primary care in China and developing countries facing the same challenge. Second, for the first time, we provide insight into the preferences of patients with T2DM regarding urban integrated primary care in China, which informs policy making about redesigning people-centered integrated health for T2DM and other chronic conditions.

The paper is subject to some limitations. First, we were unable to adopt all 21 attributes from the conceptual framework [29] in a face-to-face questionnaire survey. Hence, we could not provide a complete picture of integrated care for the respondents. To make the alternatives manageable, we identified seven priority attributes through interviews and focus group discussions. Second, as stated by the WHO, “There is no perfect combination or a ‘one size fits all’ solution” for people-centered and integrated healthcare” [35]. The same applies to patients’ preferences. The revealed preferences in this study are a combination of healthcare background, health status, health care needs, and utilization of patients, together with other factors in Nanshan district. Caution should be taken when the results and policy implications are extended to other primary care systems in China or other countries. However, the implementation of DCE to explore people’s needs and preferences for healthcare characteristics is reproducible.

## 5. Conclusions

In conclusion, this study provides important implications for policy makers, CHS managers, staff, and even T2DM patients in redesigning integrated primary care in urban China. Patients’ needs and preferences are central features of integrated care. T2DM patients value access to caring providers with rich working experience and are prepared to pay for this through out-of-pocket costs in primary care systems. People-centered interventions, such as increasing coverage by family doctors and cultivating mutual continuous relationships should be key priorities of policy development and practice guidelines. In low- and middle-income countries facing limited health resources, substantial burdens of chronic disease, and weak primary care system, it is also necessary to explore patients’ preferences and deal with the burden of disease through people-centered integrated primary care.

## 6. Data Availability

The data used to support the findings of this study are available for nonprofit research from the corresponding author upon request.

## Figures and Tables

**Figure 1 ijerph-17-00117-f001:**
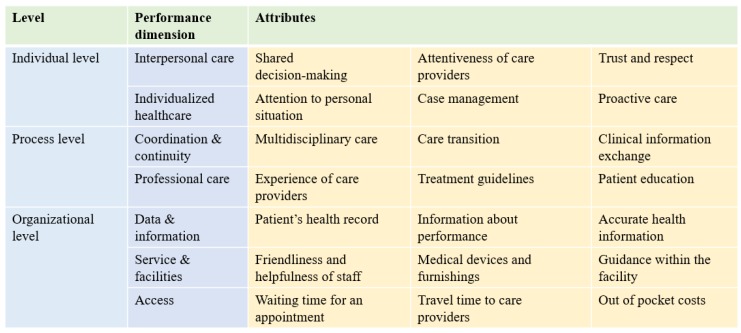
Framework of people-centered integrated care delivery (Mühlbacher, 2013).

**Table 1 ijerph-17-00117-t001:** Attributes and levels used in the discrete choice experiment.

Attribute	Description to Patients	Levels Level	
Attentiveness of care provider	…is how much time they spend helping you understand your health problems and treatment plans.	1	always given
		2	often given
		3	sometimes given
Attention to personal situation	…includes considering your work, social, and family situation in finding the most appropriate treatment for you.	1	always considered
		2	sometimes considered
		3	rarely or never considered
Multidisciplinary care	…within a single institution refers to doctors with different medical specialties working together on your treatment planning.	1	always work together
		2	sometimes work together
		3	rarely or never work together
Experience of care provider	…means the years of professional experience your care provider has.	1	>10 years of experience
		2	5–10 years of experience
		3	0–5 years of experience
Accurate health information	…refers to your ability to get accurate, reliable, and timely health and medical information.	1	very easy to get
		2	somewhat easy to get
		3	somewhat difficult to get
Friendliness and helpfulness of staff	…refers to the staff’s attitude toward you, including how often the staff is friendly and helpful when you have questions or problems.	1	always friendly and helpful
		2	often friendly and helpful
		3	sometimes friendly and helpful
Travel time to care provider	…refers to the amount of time you spend traveling to the facility of your care provider. (This does not include visits to the hospital emergency room.)	1	10 min
		2	20 min
		3	30 min
Out of pocket costs	…are expenses that you spend for additional services, prescription medicines, and for medical care treatment or testing by a doctor or another care provider per visit. These costs are in addition to the cost of health insurance.	1	¥10 per visit
		2	¥20 per visit
		3	¥30 per visit

Note: … refers to the corresponding attribute on the left column.

**Table 2 ijerph-17-00117-t002:** Characteristics of the respondents (*n* = 307).

Characteristics	No.	Percentage (%)
Age, mean (SD)	62.94 (9.93)	
Gender		
Male	148	48.20
Female	159	51.80
Marital status		
Married	258	84.04
Other marital status	49	15.96
Education		
≥junior school	163	53.10
High middle school	80	26.06
≥college	64	20.84
Employment		
Employed	69	22.47
Unemployed ^†^	238	77.53
Income (per year)		
≤¥20,000	78	25.41
¥20,000–50,000	128	41.70
¥50,000–100,000	74	24.10
≥¥100,000	27	8.79
Health insurance (Shenzhen)		
Enrolled	175	57.14
Not enrolled ^‡^	132	42.86
Diabetes complications	77	25.08
Contract with GP (%)	97	31.60
Overall health status (%)		
Very good, good, fair	105	34.20
Bad or very bad	202	65.80
Years with diabetes, mean (SD)	8.54 (7.50)	
Frequency of visits, mean (SD)	1.2 (0.99)	

^†^ Unemployed consists of retirement, unemployed for taking care of family, for illness, and for no reason. ^‡^ Not enrolled in health insurance (Shenzhen) means not enrolled in health insurance of any city or enrolled in health insurance of other cities but cannot get reimbursement of cost in Shenzhen.

**Table 3 ijerph-17-00117-t003:** Regression results and willingness to pay (*WTP*) for integrated care.

Variable ^†^	Regression Labeling	Coefficients ^‡^	Willingness to Pay ^§^	95% Confidence Interval
Out-of-pocket cost	opcost	0.24 *		
Attentiveness of care provider—always given	acp-1	0.88 *	21.89 *	14.22–29.55
Attentiveness of care provider—often given	acp-2	0.43 *	10.73 *	4.45–17.00
Attention to personal situation—always considered	aps-1	0.93 *	23.08 *	15.75–30.41
Attention to personal situation—sometimes considered	aps-2	0.56 *	13.88 *	7.47–20.29
Multidisciplinary care—always work together	mc-1	0.0005	0.01	−5.83–5.86
Multidisciplinary care—sometimes work together	mc-2	−0.43	−10.68	−16.94–4.42
Experience of care provider—over 10 years	ecp-1	1.52 *	37.72 *	26.20–49.24
Experience of care provider—5–10 years	ecp-2	0.87 *	21.44 *	13.45–29.43
Accurate health information—always easy to get	ahi-1	0.064	1.57	−3.91–7.06
Accurate health information—somewhat easy to get	ahi-2	−0.22	−5.51	−10.83–0.18
Friendliness and helpfulness of staff—always	fh-1	0.61 *	15.10 *	8.45–21.76
Friendliness and helpfulness of staff—often	fh-2	0.10 *	2.57 *	−8.43–3.30
Travel time to care providers—10 min	tt-1	1.62 *	40.18 *	29.45–50.90
Travel time to care providers—20 min	tt-2	0.72 *	17.75 *	10.98–24.52
constant	−0.53			
Number of observations	5316			
Log likelihood	−1556.72			
LR Chi^2^	571.35			
Prob > Chi^2^	0.000			
Pseudo R^2^	0.155			

^†^ Reference category is attentiveness of care provider—sometimes given, attention to personal situation—rarely or never considered, multidisciplinary care—rarely or never work together, experience of care provider—0–5 years of experience, accurate health information—somewhat difficult to get, friendliness and helpfulness of staff—sometimes friendly and helpful, travel time to care provider—30 min. ^‡^ Standard errors in parentheses. ^§^ WTP presented in ¥ per visit; $1 = ¥6.89. * Significant at 1% level.

**Table 4 ijerph-17-00117-t004:** Changes in probabilities of choosing the baseline CHS ^†^.

Change from Baseline	Change in Probability (%)	Standard Error	*p* Value
Out-of-pocket cost × 10	38.4	0.048	<0.000 *
Out-of-pocket cost × 20	19.9	0.027	<0.000 *
Attentiveness of care provider—always given	86.9	0.157	<0.000 *
Attentiveness of care provider—often given	57.4	0.145	<0.000 *
Attention to personal situation—always considered	88.6	0.161	<0.000 *
Attention to personal situation—sometimes considered	68.5	0.159	<0.000 *
Multidisciplinary care—always work together	0.9	0.203	0.996
Multidisciplinary care—sometimes work together	−88.9	0.340	0.010
Experience of care provider—over 10 years	91.7	0.154	<0.000 *
Experience of care provider—5–10 years	86.3	0.153	<0.000 *
Accurate health information—always easy to get	10.4	0.178	0.560
Accurate health information—somewhat easy to get	−41.6	0.227	0.067
Friendliness and helpfulness of staff—always	72.25	0.159	<0.000 *
Friendliness and helpfulness of staff—often	−18.4	0.227	0.419
Travel time to care providers—10 min	89.9	0.146	<0.000 *
Travel time to care providers—20 min	79.1	0.160	<0.000 *

^†^ The baseline CHS is with the lowest levels of each attribute. * Significant at 1% level.

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
