# Peer review of "How Do Type 2 Diabetes Patients Value Urban Integrated Primary Care in China? Results of a Discrete Choice Experiment"

_ijerph, 2019, doi:10.3390/ijerph17010117_

Round 1

Reviewer 1 Report

I am grateful for the opportunity to review the manuscript presented to me. I hope that the comments in the review would be helpful in deciding whether to publish the manuscript in the journal. I believe the paper is worth considering for publication, however requires  minor revision. The review is included in the original file.

Author Response

Point 1: The whole references should be improved - in my opinion only partially in line with the IJERPH guidelines. I have marked only an example of inconsistency in the way of writing quoted items.

Response 1: We would like to thank the reviewer for this advice. All the references have been improved according to the IJERPH guidelines.

Reviewer 2 Report

The study provides valuable insights into patient preferences that if taken into account would make integrated primary healthcare more accessible and appealing to the T2DM patient population in the said district in China. While the authors aptly suggest that the results of this study cannot be extrapolated to other regions, some of the key findings of this study would be informative for applying across different regions. Through the importance of the years of experience of the doctors, we can glean the trust that the patients place in the doctors. However, it would have been interesting to know patient responses directly to the question of trust in the doctor/care staff's ability to diagnose and deliver care. While not directly quantifiable, measures to improve the image of the care community amongst patients could also serve to compensate for the lack of years of experience. 

Minor corrections:

Abstract:

Give full from of the abbreviation T2DM in the abstract as it is the first instance of using the abbreviation.

Line 37:

Full form of DALYs and YLDSs?

Line 73: Format reference numbers [18-20]

Author Response

Minor corrections:

Abstract: Give full from of the abbreviation T2DM in the abstract as it is the first instance of using the abbreviation.

Line 37: Full form of DALYs and YLDSs?

Line 73: Format reference numbers [18-20]

Response :

We would like to thank the reviewer for these corrections.

Full form of T2DM, DALYs and YLDs have been added, and brackets have been added for references 18-20 in line 73.

Reviewer 3 Report

Comments and Suggestions for Authors

The manuscript presented for review is very interesting and I recommend the article for publication in International Journal of Environmental Research and Public Health. In my opinion, the paper is important for the world scientific society. However, manuscript should be improved. Please note and address the following comments.

Comment 1: Introduction

First, the introduction does not demonstrate what the healthcare system in China  looks like, as well as how it relates to the systems functioning in the world. Is it specific or similar to others systems? In my opinion, this should be improved so that every reader understands the intentions of the authors.

Comment 2: Material and methods

In table 1 is 8 attributes on 24 levels used in the Discrete Choice Experiment, but in manuscript in chapter Results (lines 184-185) authors wrote about 8 or 9 choices. I am not sure if I well understand this experiment, what kind of survey was to be completed by patients. Did everyone get 3 questionnaires?

Citation:

Lines 184-185: “The final sample used in the analysis comprised of 307 respondents and 5318 observations, with each respondent providing response to 8 or 9 choices.

Lines 129-130: "To avoid overloading the respondents, we randomly divided the 26 choice sets 129 into 3 different questionnaires".

Lines 121-124: "We used 27 alternatives, which is more manageable, using a fractional factorial experimental design [31]. The alternative with the most attributes at middle level was selected as a comparator and the other 26 alternatives were compared with it, to make the choices easier to understand for respondents [32].”

Comment 3: Material and methods

Lines 135: 2.2.3. Sampling and data collection

Lines 136-139: In manuscript is “The first stage sampled CHSs, and the second stage sampled T2DM patients. 16 of 80 community health stations (20%) were selected by systematic sampling. In each CHS, 20 patients with T2DM were randomly selected from the list of residents who registered with GP in the CHS”.

Taking into consideration this assumption, there should be 320 patients in the experiment, and there are 307. What happened to the other questionnaires?

Comment 4: Conclusion

What are the practical and theoretical implications of the research?

In my opinion, in chapter Conclusion is lack an information on how the obtained results can be related to other countries. Is it possible to suggest any solutions for improving the healthcare system in China based on the obtained data?

Author Response

Point 1: First, the introduction does not demonstrate what the healthcare system in China looks like, as well as how it relates to the systems functioning in the world. Is it specific or similar to others systems? In my opinion, this should be improved so that every reader understands the intentions of the authors.

Response 1: We would like to thank the reviewer for this advice.

The setting of this study is Chinese primary care system. As we demonstrated in the first paragraph of the Introduction part, “Yet, the Chinese health system has been characterized by a disease focused hospital-centered approach with a weak primary care system since the 1980s”. And we introduced solutions taken in recent years to strengthening diabetes-related care delivery through Chinese primary care system.

In the second paragraph of the Introduction part, we demonstrated that “In the last decade, integrated care for diabetes and other chronic conditions has been recognized as a means to meet the needs of this population by promoting care coordination, improving the quality of care and generating better patients’ outcomes in developed countries.” It means that primary care systems in developed countries also need integrated care, which is similar to Chinese primary care systems.

Integrated care has been provided in some developed countries based on patients’ preferences. But, we have no idea about patients’ preferences on integrated care in Chinese primary care systems. Therefore, the aim of this study is to elicit the preference regarding the use of urban integrated primary care in China using a DCE and interviews with T2DM patients.

Point 2: In table 1 is 8 attributes on 24 levels used in the Discrete Choice Experiment, but in manuscript in chapter Results (lines 184-185) authors wrote about 8 or 9 choices. I am not sure if I well understand this experiment, what kind of survey was to be completed by patients. Did everyone get 3 questionnaires?

Response 2: Sorry we did not interpret clearly about our design.

A full factorial design of 8 attributes (with 3 levels each) generates 38=6567 alternatives. Using a fractional factorial experimental design, 27 alternatives were selected. The alternative with the most attributes at middle level was selected as a comparator and the other 26 alternatives were compared with it, generating 26 choice sets. Then, we randomly divided the 26 choice sets into 3 different questionnaires, with two questionnaires containing 9 choice sets each and one questionnaire containing 8 choice sets. Each respondent just answers one of the 3 questionnaires.

Point 3: In manuscript is “The first stage sampled CHSs, and the second stage sampled T2DM patients. 16 of 80 community health stations (20%) were selected by systematic sampling. In each CHS, 20 patients with T2DM were randomly selected from the list of residents who registered with GP in the CHS”. Taking into consideration this assumption, there should be 320 patients in the experiment, and there are 307. What happened to the other questionnaires?

Response 3: 13 respondents refused to participate in our investigation. We interpreted in line 188 as “The response rate was 95.9%”.

Point 4: What are the practical and theoretical implications of the research? In my opinion, in chapter Conclusion is lack an information on how the obtained results can be related to other countries. Is it possible to suggest any solutions for improving the healthcare system in China based on the obtained data?

Response 4: The most important implication of our study is to inform policy making about redesigning people-centered integrated health care for T2DM patients in China. Solutions such as hiring staff with a few years working experience, and encouraging endocrinologists with rich experience from district hospitals to set up clinics regularly in CHSs can help improving the primary health care system in China. We did realize that due to differences of health care system, caution should be taken when the results and policy implications are extended to other countries. We have acknowledged this in “Strengths and Limitations”.